# Did the Introduction of Biosimilars Influence Their Prices and Utilization? The Case of Biologic Disease Modifying Antirheumatic Drugs (bDMARD) in Bulgaria

**DOI:** 10.3390/ph14010064

**Published:** 2021-01-14

**Authors:** Konstantin Tachkov, Zornitsa Mitkova, Vladimira Boyadzieva, Guenka Petrova

**Affiliations:** 1Department of Organisation and Economy of Pharmacy, Faculty of Pharmacy, Medical University of Sofia, 1000 Sofia, Bulgaria; tachkov@outlook.com (K.T.); zmitkova@pharmfac.mu-sofia.bg (Z.M.); 2Department of Rheumatology, Faculty of Medicine, Medical University of Sofia, 1612 Sofia, Bulgaria; vladimira.boyadzhieva@gmail.com

**Keywords:** biosimilars, pricing, reimbursement, utilization

## Abstract

The aim of this study is to evaluate the effect of the introduction of biosimilars in Bulgaria on the prices and utilization of biologic disease modifying antirheumatic drugs (bDMARD). It is a combined qualitative and quantitative analysis of time of entry of biosimilars on the national market and the respective changes in the prices and utilization during 2015–2020. We found 58 biosimilars for 16 reference products authorized for sale on the European market by the end of 2019, but for 2 of the reference products biosimilars were not found on the national market. Only inflammatory joint disease had more than one biosimilar molecule indicated for therapy. Prices of the observed bDMARD decreased by 17% down to 48%. We noted significant price decreases upon biosimilar entrance onto the market. In total, the reimbursed expenditures for the whole therapeutic group steadily increased from 72 to 99 million BGN. Utilization changed from to 0.5868 to 2.7215 defined daily dose (DDD)/1000inh/day. Our study shows that the entrance of biosimilars in the country is relatively slow because only half of the biosimilars authorized in Europe are reimbursed nationally. Introduction of biosimilars decreases the prices and changes the utilization significantly but other factors might also contribute to this.

## 1. Introduction

It is largely well-established, as supported by evidence, that after the introduction of generic medicines the price of originals decreases, allowing for an increase in medicines utilization [1,2,3,4]. This is mostly valid for the synthetic medicines where the criteria for essential similarity between the originator and off-patented versions are scientifically and regulatory established [5]. Generic medicines benefit the market by offering equally high-quality treatment as originator medicines but at much lower prices [6,7]. Based on the essential similarity of medicines, countries introduce a variety of measures to stimulate generic medicines manufacturing, prescribing, dispensing, and utilization in the society [8,9]. Those measures are described as generic medicines policy [10,11]. Generic medicines policy has been promoted by the World Health Organization for many years with the main goal of encouraging governments to introduce it as part of their national drug policy [12,13]. A core element of the generic medicines policy is a list of essential drugs comprising the most widely used by the majority of people and medicines for a large number of diseases [14]. The aim of introducing generic incentives is to foster competition, decrease prices, and enlarge the utilization of essential medicines, thus, covering the needs of the majority of the population [15,16,17].

Biological medicines encompass a wide group of therapeutic agents that are manufactured through living organisms and include monoclonal antibodies, peptides (e.g., insulin), vaccines, blood products, RNA targeting therapies, and gene and cellular therapies [18]. A biosimilar medicine is a biological medicine that is similar to another biological medicine that has already been authorized for use [19,20,21]. Biosimilars have a number of important differences from generic small-molecule drugs, including manufacturing processes that are unique from their reference products (i.e., originators) [20]. There is considerable debate, still, in the scientific community about the safety and interchangeability of biological products [21]. Authors consider that the availability of biosimilars might provide an opportunity to lower health care expenditures as a result of the inherent price competition with their reference product [22]. This is due to the fact that biological products are rising as a proportion of drug expenditures globally [23]. There are estimates that over 30% of all drug spending in Europe is on biological medicines and out of them 1.5% are for biosimilars. There has been an increase by 3.4% over the last five years for all biologic medicines, and by 1.2% since 2014 for biosimilars. By the end of 2018, 16 biological molecules have had biosimilar products introduced in Europe, meaning that there is a possibility to enhance biosimilars competition with reference biological products. In countries where there is no officially introduced generic medicines policy, we can expect obstacles towards the market penetration, prescribing, and competition in the field of biological products [24,25]. This stimulated our interest towards the topic.

The aim of this study is to evaluate the effect of the introduction of biosimilars in Bulgaria on the prices and utilization of biologic disease modifying antirheumatic drugs (bDMARD) for inflammatory joint diseases therapy.

## 2. Results

### 2.1. Availability of Biosimilars in the Reimbursement Drug List

We found 58 biosimilars for 16 reference products authorised for sale on the European market by the end of 2019, but for 2 of the reference products (insulin lispro and enoxaparin) biosimilars were not found on the national market. The national market included 14 reference products for which 29 biosimilars are reimbursed (Table 1).

Fifteen biosimilars are reimbursed for the outpatient practice. Adalimumab, infliximab, etanercept, and rituximab are biologic disease modifying antirheumatic drugs (bDMARD) indicated for inflammatory joint diseases therapy for which nine biosimilars were found. The rest of the INNs for outpatient practice possess only one biosimilar reimbursed in the country, except epoetin alfa with two biosimilar (Table 1).

Trastuzumab, bevacizumab, filgrastim, pegfilgrastim, and follitropin alfa are reimbursed for inpatient practice but for different therapeutic areas and we could not compare them at a therapeutic level. Fourteen biosimilars are authorized for those INNs (Table 1).

From Table 1, it is also evident that the time period for biosimilar entry on the national reimbursed market after its marketing authorization in Europe varies from two months (infliximab first biosimilar) to nine years (bevacizumab biosimilar). It is also evident that a total six biosimilars were present in the reimbursed practice for a limited time period which were later excluded from the Positive Drug List probably due to marketing authorization holder request.

Reviewing the changes in the authorized for sale and reimbursed indications of inflammatory joint disease therapy, we found the following. Infliximab was the first reimbursed by the National Health Insurance Fund bDMARD for the indication of RA therapy, subsequently AS was added, as well as PSA as an indication. All of the indications were approved by the NCPR prior to the observed period 2015–2019. During the observed period, entrance of an infliximab biosimilar to the reimbursement practice allowed for reimbursement of all aforementioned indications immediately upon receiving approval. This was the case for other biosimilars for inflammatory joint diseases therapy as well.

### 2.2. Changes in Prices and Utilization of Anti-Inflammatory Joint Diseases Medicines

During 2015–2020, a total of nine biosimilar products for four INNs were found available on the reimbursement drug list within the group of anti-inflammatory joint diseases (Table 2).

In 2015, infliximab had only two biosimilar alternatives available, with a third being introduced in 2019; however, this inclusion was of the originator of infliximab. Its price was influenced by the already established reference price per DDD of the corresponding biosimilars as per the active legislation requiring the reimbursing of the lowest price per DDD. The price of infliximab dropped down by nearly 41% during the period and the highest decrease was observed when the third product was included in the list. One biosimilar for etanercept entered the reimbursement practice leading to almost a 75% decrease in the reference price per DDD at the moment of entrance and total 49% price decrease during the whole period observed. Adalimumab appears to be the most competitive INN with four biosimilars introduced during the period and nearly double the decrease in the price. Originator for tocilizumab added one new dosage form leading to a small decrease in reference price per DDD. For rituximab, we found one included biosimilar and the price dropped down by 1.3 BGN per DDD. This dosage form was subsequently excluded, leading to an increase in reference price with 0.27BGN (Table 2).

Regarding the changes in prices, we observed a decrease by nearly 50% for INNs where biosimilars were introduced and with 5–21% for INNs where there was no biosimilar competitor available prior to introduction (Table 2). The decrease in the prices of the other INNs where there is no biosimilars could be explained with the regular price revision. If there is a price decrease in the reference countries it immediately affects the prices on the national market. For the period 2015–2020, all changes in prices were found to be statistically significant (*p* < 0.001). The inclusion of baricitinib resulted in a nonsignificant change in prices for the period 2019–2020 (*p* = 0.1326).

Reimbursed expenditures increased for almost all INNs in 2019 in comparison with 2015 (Table 3). Only for infliximab we noted a decrease in expenditures by 49%, which could be attributed both partly to the decreasing prices, and partly due to the entrance of new therapeutic and biosimilar competitors. Similar is the situation with adalimumab, whereby until 2018 reimbursed expenditures increased and upon the introduction of biosimilars it started to decrease. We can assume that the price decrease leads also to decrease in the reimbursed expenditures.

In total, the reimbursed expenditures for the whole therapeutic group steadily increased from 72 to 99 million BGN by the end of the observed period (nearly 36–45.9 million Euro)—Figure 1. Variations in the percent change of total reimbursed expenditures was noted between 2017 in comparison to 2016 (*p* < 0.05) when the increase is less than between 2016 and 2015 (*p* > 0.05). In 2019, we observed a decrease in total expenditures with 2.83%, which was nonsignificant. However, the change in expenditures between 2017 and 2018 were significant (*p* < 0.05), and seem to be largely influenced by the increased expenditures for secukinimab and rituximab.

Utilization in DDD/1000inh/day is stable for most INNs, with a smooth increase except for rituximab and adalimumab (Figure 2) for which a significant increase is observed.

As a whole, during 2015–2019 the utilization in DDD/1000inh/day increased from 0.657 to 2.395. The increase in utilization was found to be significant (*p* < 0.001). Adalimumab is definitely a leader in utilization in DDD/1000inh/day accounting for 32% to 21% of total utilization during the period (Table 4). For rituximab we noted a tremendous increase in utilization in 2018 when the first biosimilar entered the market.

What is worth noting, however, is that all changes in utilization were found to be significant, even those between 2018 and 2019 (*p* < 0.05), despite the changes in total NHIF expenditures for the same time period being nonsignificant. This seems to indicate that introduction of biosimilars and the implementation of cost-containment measures is able to control for an increase in expenditures, and allow for increase in utilization of these medicines.

## 3. Discussion

To the best of our knowledge, this is the first national study exploring the entrance of biosimilars on the national market and their influence on the reimbursed prices and utilization of a particular therapeutic group in DDD/1000inh/day. There are two other national studies focusing on biosimilars [26,27]. The first one compared the prices of biological products for rheumatoid arthritis therapy, and found that manufacturer prices of reference biological product and biosimilars shows 36% difference for etanercept, 39% for rituximab, and 31% for infliximab, while at retail level the differences are 11%, 86%, and 143%, respectively [28]. It does not explore their reimbursement prices and utilization, but only officially published manufacturer and retail prices. Authors noted this as a limitation of the study. The second article explores the access to biotechnological drugs for rare diseases and found that they comprise a high proportion of pharmaceutical expenditures in the reimbursed biotechnological medicinal products market [29].

Similar international comparisons reviewed the requirements for reimbursement of biosimilars and compared the reimbursement status, market share, and reimbursement costs of biosimilars in Bulgaria, the Czech Republic, Croatia, Estonia, Hungary, Latvia, Lithuania, Poland, Slovakia, and Romania during 2016–2017, using a questionnaire, focusing mostly on the regulatory requirements for the pricing and reimbursement of biosimilars for each country [30]. Authors pointed out that the total expenditure on the reimbursement of biologic drugs in the CEE countries was 397,097,152 EUR in 2014 and 411,433,628 EUR in 2015, but the data for Bulgaria was scarce.

None of the national studies explore the entrance of all biosimilars. Our research found that almost half of all authorized by EMA biosimilars are available on the market but only in one therapeutic group could we establish price competition. There are still many biosimilars that are not available. In addition to this slow penetration, the time for entrance is also variable. For the earlier biosimilars it was extremely long (nine years for the epoetin alfa) but in recent years, the time of inclusion has become faster; in some cases, as quickly as two months, which indicated progress in marketing penetration. This might be also due to the fact that in the area of bDMARD for inflammatory joint diseases therapy not only is biosimilar competition increasing, but also therapeutic competition, and the range of improved indications has expanded to cover other forms of arthritis. The first bDMARD (infliximab) for joint diseases therapy were positioned for rheumatoid arthritis in 2000, subsequently the indications increased to allow for other types of inflammatory joint diseases (RA, PSA, AS). In 2003, the indication AS was added, and in 2004 the indication of psoriatic arthritis (PSA) was approved. Despite the approval received by EMA for the treatment of RA in 2000 for etanercept and 2003 for adalimumab, the NHIF added both bDMARDS with a significant delay at the end of 2009, but for the three inflammatory joint diseases (RA, PSA, AS). In 2010, the NHIF included in the list two new molecules: anti-IL-6-tocilizumab and anti-CD20-rituximab with indication RA (rituximab received approval also for Wegener’s disease in 2015). One year later, certolizumab pegol was included in the therapeutic arsenal, approved by the NHIF, for the indication RA, and in 2015 for the other two inflammatory joint diseases. In 2012, golimumab received approval for the three diagnoses and ustekinumab a year later, but only for the indication of psoriatic arthritis. The last two bDMARDs received approval in 2017 for secukinumab and in 2019 for ixekizumab. A new group of medicines-target synthetic DMARDS has entered widely in the practice of rheumatologists in 2018. Tofacitinib was the first approved by the NHIF in March 2018, followed by baricitinib in 2019. To date, no biosimilar products of these have been presented but we found that their entrance changes the utilization in the group as a whole.

It is also important to note that biosimilars entrance is delayed also by the market exclusivity practices of the pharmaceutical companies [31].

A limitation of our study is that we focused only on the therapeutic group for outpatient practice, because the reimbursed prices of medicines for hospitals are an object of tenders and all of them are also subject of confidential rebate negotiation so the real market price could not be established.

Regarding the prices, we confirmed the hypothesis that the biosimilars decrease the prices of biological product even at the moment of their entrance in the reimbursement system. The prices are highly competitive and in comparison with the INNs, where there is no biosimilars, prices are falling down at twice the rate [27]. The pricing policy in Bulgaria is oriented towards lower costs and lower prices. External reference pricing is applied for price approval and lowest ex-manufacturing price is used out of 10 reference countries. After the reimbursement approval, the lowest price per DDD is used as a reference price for reimbursement within the INN. The fact that the years with the most included biosimilars (2018–2019) had a nonsignificant change in total expenditures indicates that these cost-containment measures are effective.

We also confirm that the entrance of biosimilars influences the utilization in a positive direction, except for infliximab, with significant changes being observed for all INNs. The decrease in utilization of infliximab could be attributed to the constantly lowering prices and entrance of new bDMARDS within the group. This is probably influenced by adalimumab who is the leader in the group. Adalimumab is one of the most commonly prescribed blockers of TNFa due to its well-established long-term safety profile [28], tolerability, and effectiveness compared to other bDMARDS [32]. It is one of the first three bDMARDS approved for treatment by the NHIF with 18 indications to date. Recent studies reveal that adalimumab is one of the most prescribed biologics in the United States after an analysis of the treatment of 40,373 RA patients [33].

A study of the utilization of biosimilars was conducted in Korea, where authors found an increasing market share for infliximab biosimilars at over 30%, while rituximab and trastuzumab had a share of 12.89% and 13.93%, respectively [34]. They also found savings over six years after the biosimilar entry to the market. A similar study explores the utilization of infliximab and filgrastim on the US market and it was one of the first matching the importance of biosimilar products [35]. The cost savings are considered as benefits from the introduction of biosimilars [36]. Other authors also prove that biosimilars not only decrease the prices but also increase the utilization but still there are concerns for their interchangeability [37,38].

The other study discussed the market drivers for biosimilars [39]. The authors confirm that there is a correlation between the biosimilar penetration and price decrease. They consider that incentive policies to enhance uptake remain an important driver of biosimilar penetration. The only incentive that is available at the moment in Bulgaria is that the price of biosimilar should be no more than 80% of the price of originator, but it was introduced in the legislation just in 2018 so it does not affect the whole period studied [40]. Therefore, we could not consider that this change in regulation is influencing the price decrease during the whole period.

## 4. Materials and Methods

The study utilized a combined quantitative and qualitative analysis of time of entry of biosimilars on the national market and the respective changes in the prices and utilization of biologic disease modifying antirheumatic drugs (bDMARD).

### 4.1. Qualitative Analysis

The qualitative analysis included comparisons of market entry of biosimilars—their time of approval and entry onto the Bulgarian market. Information regarding all authorized biosimilars and approval of their new indications till the end of 2019 was taken from the EMA webpage [41]. Subsequently, the Internet page of the National Council of Prices and Reimbursement (NCPR) [42] was searched for reimbursed biosimilars up until the end of 2020.

The availability was presented as the number of biosimilars per international nonproprietary name (INN), authorized by EMA and available on the European market, which was then compared to the date of product entry into the reimbursement list on the national market.

### 4.2. Quantitative Analysis

After the qualitative analysis, we selected a single therapeutic group-biologic disease modifying antirheumatic drugs (bDMARs) for further analysis. The choice was based upon the fact that this therapeutic group had the largest number of reimbursed biological and biosimilar products for the longest duration of time. Under inflammatory joint diseases we encompass rheumatoid arthritis (RA), psoriatic arthritis (PSA), and ankylosing spondylitis (AS) because those are the most often reimbursed diagnoses.

Two data sources were accessed for the quantitative analysis—the National Health Insurance Fund (NHIF) and the NCPR. From the NHIF we extracted data on the reimbursed expenditures for the period 2015–2019 of bDMARD for inflammatory joint disease medicines. The changes in expenditures for every year are presented in national currency (BGN) at the exchange rate of 1 BGN = 0.95 Euro. The exchange rate of BGN to Euro in Bulgaria has been fixed since 1997.

The NCPR database was accessed retrospectively to follow changes in medicine reference prices per defined daily dose (DDD) and per INN throughout the period 2015–2020. The reference price per DDD is the lowest reimbursed price per DDD.

Utilization of the medicines for inflammatory joint diseases was analyzed in monetary units and in DDD/1000inh/day by using the WHO formula ((Sales data/DDD/number of inhabitants/365) × 1000) [43].

### 4.3. Statistical Analysis

Friedman’s variant of ANOVA was applied for all years for which data was available to follow the changes in prices and utilization. Where a new medicine was introduced, and data was available only for two years, a Wilcoxon nonparametric analysis was applied to analyze the changes in therapy for all biologics. *p*-values of less than 0.05 were considered statistically significant. The software package MedCalc version 19.6 was used.

## 5. Conclusions

Our study shows that the entrance of biosimilars in the country is relatively slow because only half of the authorized biosimilars in Europe are reimbursed. Introduction of biosimilars decreases the prices and changes the utilization significantly but other factors might also contribute to this.

## Figures and Tables

**Figure 1 pharmaceuticals-14-00064-f001:**
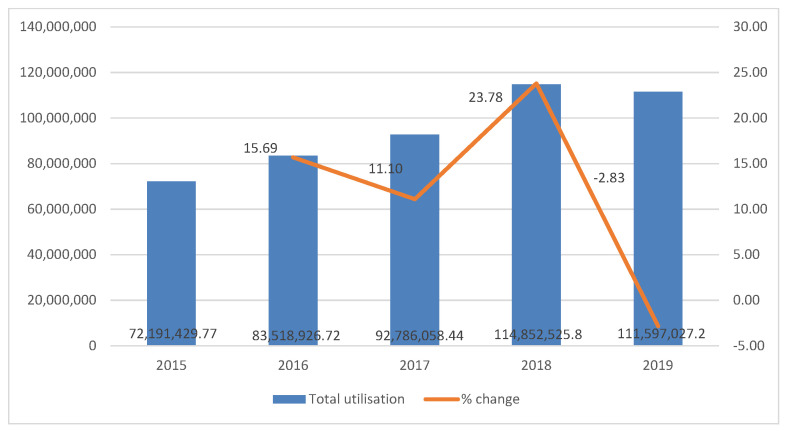
Total reimbursed expenditures and % change every year.

**Figure 2 pharmaceuticals-14-00064-f002:**
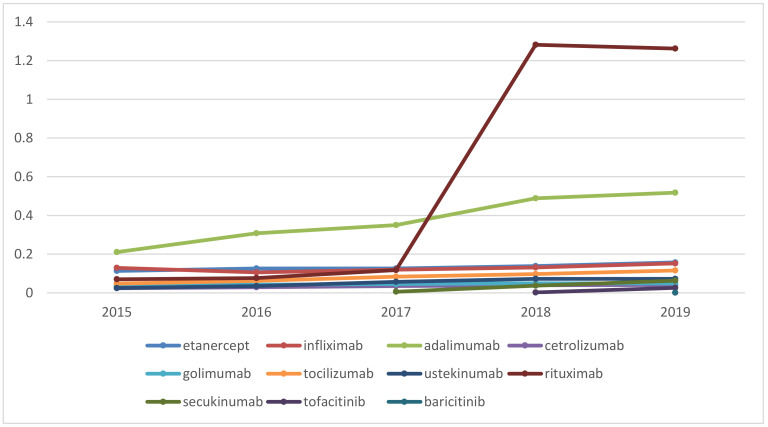
Utilization in DDD/1000inh/day.

**Table 1 pharmaceuticals-14-00064-t001:** Available biosimilar in the reimbursement practice and date of entrance.

INN	Authorized Biosimilars in Europe (*n*)	Biosimilars Available on the National Market (*n*)	Authorisation Date in Europe	Date of Inclusions into the Positive Drug List (PDL)	Time Lag
**Outpatient Practice**					
insulin glargine	2	1	8/09/2014	24/08/2015	11 months
adalimumab	8	1	20/03/2017	22/10/2018	1 year 5 months
		1	15/09/2018	22/03/2019	6 months
		1	25/07/2018	18/03/2019	6 months
		1	16/09/2018	20/02/2020	2 year 6 months
infliximab	4	1	9/09/2013	27/11/2013	2 months
		1	8/09/2013	27/11/2013	2 months
		1	17/05/2018	28/03/2019	10 months
etanercept	2	1	22/06/2017	19/06/2020	3 year
rituximab	5	1	16/02/2017	30/03/2018 2/11/2020 (excluded)	11 months 2 year
epoetin alfa	3	1	26/08/2007	20.03.2013 2.10.2014 (excluded)	4 year 6 months 1 year 7 months
		1	22/08/2007	26/09/2016	9 year
epoetin zeta	3	1	17/12/2007	15/06/2012	4 year 5 months
teriparatide	3	1	10/01/2017	8/11/2019 2/04/2020 (excluded)	1.9 year 6 months
somatropin	1	1	11/04/2006	3/12/2011	4 year 8 months
**Inpatient Practice**					
follitropin alfa	2	1	25/03/2014	01/03/2015	1 year
filgrastim	6	1	16/09/2014	01/02/2018	3 year 5 months
		1	6/06/2010	22/08/2011 8/09/2012 (excluded)	1 year 2 months 1 year
		1	14/09/2008	21/08/2012 16/06/2014 (excluded)	4 year 2 year
		1	5/02/2009	19/09/2011	2 year 7 months
pegfilgrastim	6	1	25/04/2019	27/03/2020	11 months
		1	20/09/2018	01/03/2019	5 months
		1	21/11/2018	01/03/2019	4 months
trastuzumab	5	1	7/02/2018	12/10/2018 2/11/2020 (excluded)	8 months 2 year
		1	15/05/2018	2/02/2019	9 months
		1	11/12/2018	13/03/2019	3 months
		1	14/11/2017	21/02/2020	2 year 3 months
		1	25/07/2018	13/03/2019	6 months
bevacizumam	2	1	14/01/2018	28/09/2020	1 year 9 months

**Table 2 pharmaceuticals-14-00064-t002:** Reference price per define daily dose (DDD) for anti-inflammatory joint diseases medicines (BGN).

INN	2015	2016	2017	2018	2019	2020	% Change (2020 to 2015)	Fisher Test
etanercept	67.08	60.49	58.69	54.55	53.75	34.39 *	48.72	*p* < 0.0001
infliximab	30.61 *,*	28.68	27.018	27.02	17.92 *	17.92	41.47	
adalimumab	72.39	66.08	66.08	50.01*	38.13 *	38.13 *,*	47.33	
cetrolizumab	63.67	57.29	57.29	53.10	53.11	50.03	21.43	
golimumab	65.56	65.24	62.05	57.50	57.50	53.34	18.63	
ustekinumab	73.70	73.24	65.53	60.65	60.65	60.65	17.71	
tocilizumab	71.21	70.91	60.58	60.58	59.22	54.69	23.19	
rituximab	4.97	4.97	4.94	3.65 *	3.13	3.39 ↓	31.66	
secukinumab			81.04	80.88	73.08	72,58	10.44	
tofacitinib				55.51	52.94	52.94	4.63	
baricitinib					77.16	77.16	0%	

**Legend:** * biosimilar included; ↓ biosimilar excluded.

**Table 3 pharmaceuticals-14-00064-t003:** Reimbursed expenditures in monetary units (in millions of BGN).

INN	2015	2016	2017	2018	2019	% Change (2019 to 2015)	Fisher Test
Etanercept	15.87	15.78	15.27	15.47	17.19	7.69	*p* < 0.0001
Infliximab	8.25	6.30	6.65	7.24	5.54	−48.77	
Adalimumab	31.86	42.24	47.70	49.99	40.08	20.51	
Cetrolizumab	3.19	3.45	4.17	4.32	4.43	27.98	
Golimumab	4.98	5.71	5.82	5.85	5.71	12.89	
Tocilizumab	7.15	9.06	10.46	12.00	13.91	48.58	
Rituximab	0.89	0.97	1.49	11.96	10.06	91.12	
Secukinimab			1.23	7.72	11.74	89.56	
Tofacitinib				0.28	2.80	89.87	
Baricitinib					0.12	0	

**Table 4 pharmaceuticals-14-00064-t004:** Share of utilization in DDD/1000inh/day.

	2015	% of Overall Utilization	2016	% of Overall Utilization	2017	% of Overall Utilization	2018	% of Overall Utilization	2019	% of Overall Utilization
etanercept	0.1133	17.25	0.1258	16.04	0.1263	13.41	0.1387	5.93	0.1575	6.41
infliximab	0.129	19.63	0.1059	13.50	0.1197	12.71	0.1312	5.61	0.1524	6.20
adalimmab	0.2107	32.07	0.3082	39.29	0.3506	37.22	0.489	20.91	0.5178	21.07
cetrolizumab	0.0239	3.64	0.0291	3.71	0.0353	3.75	0.0398	1.70	0.041	1.67
golimumab	0.0363	5.53	0.0423	5.39	0.0456	4.84	0.0497	2.13	0.0489	1.99
tocilizumab	0.0481	7.32	0.0616	7.85	0.0839	8.91	0.0969	4.14	0.1157	4.71
rituximab	0.0254	36.13	0.0352	4.49	0.0567	6.02	0.072	3.08	0.0726	2.95
secukinumab	0.0703	10.70	0.0764	9.74	0.1179	12.52	1.2817	54.80	1.2619	51.35
ustekinumab					0.0059	0.63	0.0374	1.60	0.0629	2.56
tofacitinib							0.0024	0.10	0.026	1.06
baricitinib									0.0007	0.03
**total utilization**	0.66	132.26	0.78	100.00	0.94	100.00	2.34	100.00	2.46	100.00

## Data Availability

Data sharing not applicable.

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
