# Peer review of "Did the Introduction of Biosimilars Influence Their Prices and Utilization? The Case of Biologic Disease Modifying Antirheumatic Drugs (bDMARD) in Bulgaria"

_pharmaceuticals, 2021, doi:10.3390/ph14010064_

Round 1

Reviewer 1 Report

The research aimed to evaluate how the introduction of biosimilars influence their prices and utilization in Bulgaria for bDMARDs.

It is a valuable summary and I hope you find the suggested changes helpful:

Title need insertion of "?" in place of "-" and a "s" for "bDMARDs": "Did the introduction of biosimilars influence their prices and utilization? The case of Bulgaria for biologic disease modifying antirheumatic drugs (bDMARDs)"

The abbreviations must be declare at their first use and then use only the abbreviation.

You need to review all these abbreviations:

-line 80:National Council of Prices and Reimbursement (NCPR) not at line 93

-line 82: you use INN, but explain at line 98

-line 90: PSA is psoriatic arthritis and you use PA at line 132

-line 93: NHIF is explained and then you use National Health Insurance Fund at line 232 and 255

In table 1:you use yr and years, explain what is PDL and change the title of the column 2 and 3: "Authorized biosimilars in Europe (n)" and "Biosimilars available on the national market (n)"

Line 86: I think it is "After the qualitative analysis..." and at line 92: "for the quantitative analysis.."

The quality of figures and tables is satisfactory.

The methods are sufficiently documented to allow replication studies.

For statistical analysis, you must specify the level of p-value that was considered statistically significant and the statistical software you used.

You included secukinumab in Figure 2, but not in the Table 4. Why?

The reference list covers the relevant literature adequately, in an unbiased manner, from the last years.

Author Response

Dear Editor and Reviewer 

thank you for your valuabe comments. Attached is the file with point by poinbt answers to all suggestions by both reviewers because some were overlapping.

Wish you healthy and prosperios New Year.

Reviewer 2 Report

Thanks for the opportunity to review this manuscript. It reports on a well-designed and conducted study that addresses an important topic: the uptake of biosimilars and its impact on both price and utilization of biologics.

I recommend acceptance subject to addressing the following comments:

Title:

  • I suggest changing to be grammatically correct, to:

Did the introduction of biosimilars influence their prices and utilization? The case of biologic disease modifying antirheumatic drugs (bDMARD) in Bulgaria.

Introduction:

  • Line 34: please delete “with multiple evidences” as grammatically incorrect or change to “ as supported by evidence”
  • Line 40: generices should be singular “generic”
  • Line 44: the sentence “Core element in the generic medicines policy are the essential drugs as mostly widely used medicines by the prevailing part of the population for the majority of the diseases” is not very clear. Do you mean: “Core element in the generic medicines policy is a list of the essential drugs that are most widely used medicines by the majority of the population for a large number of diseases”??
  • Line 54: “Therefore, essential similarity is not yet proven for biosimilar medicines” is not strictly correct is essential similarity is unlikely to be proven at all. Perhaps rephrase to make this clear.
  • Line 56 and 57: “There are still lots of debates in the scientific society” should be corrected as follows: “There is considerable debate, still, in the scientific community”
  • It would be useful to add some information about the market of biologics in Bulgaria specifically as the introduction is focused on Europe in general.

Materials and methods:

  • Line 73: “It is a” should be changed to “The study utilized a”
  • Line 89: delete the from “the rheumatoid”
  • Line 96: You have used a constant exchange rate of BGN 1 =0.95 Euro for all 5 years of the period 2015-2020 (or is it to 2019?). Is this correct? How did you decide on this exchange rate? Please clarify.

Results:

  • Line 23: space to be added between 14 and biosimilars
  • Line 126: “6 biosimilar” should be “6 biosimilars
  • Table 3 could be better presented by expressing the numbers in millions of BGN

Author Response

Dear Editor and reviewer,

Thank you for your valuabe comments and suggestions.

Attached please find the point by point aswer to both reviewers because some were overlapping.

Wish you health and prosperious New Year.
